# HCS-Splice: A High-Content Screening Method to Advance the Discovery of RNA Splicing-Modulating Therapeutics

**DOI:** 10.3390/cells12151959

**Published:** 2023-07-28

**Authors:** Giuseppina Covello, Kavitha Siva, Valentina Adami, Michela Alessandra Denti

**Affiliations:** 1RNA Biology and Biotechnology Laboratory, Department of Cellular, Computational and Integrative Biology—CIBIO, University of Trento, 38123 Trento, Italy; kavithas@trilliummedvillage.com; 2High Throughput Screening and Validation Core Facility (HTS), Department of Cellular, Computational and Integrative Biology—CIBIO, University of Trento, 38123 Trento, Italy; valentina.adami@unitn.it

**Keywords:** alternative splicing, two-colour (GFP/RFP) fluorescent reporter, *MAPT*, exon-skipping, FTDP-17, high content screening, siRNA, nucleic acids therapeutics, drug discovery

## Abstract

Nucleic acid therapeutics have demonstrated an impressive acceleration in recent years. They work through multiple mechanisms of action, including the downregulation of gene expression and the modulation of RNA splicing. While several drugs based on the former mechanism have been approved, few target the latter, despite the promise of RNA splicing modulation. To improve our ability to discover novel RNA splicing-modulating therapies, we developed HCS-Splice, a robust cell-based High-Content Screening (HCS) assay. By implementing the use of a two-colour (GFP/RFP) fluorescent splicing reporter plasmid, we developed a versatile, effective, rapid, and robust high-throughput strategy for the identification of potent splicing-modulating molecules. The HCS-Splice strategy can also be used to functionally confirm splicing mutations in human genetic disorders or to screen drug candidates. As a proof-of-concept, we introduced a dementia-related splice-switching mutation in the Microtubule-Associated Protein Tau (*MAPT*) exon 10 splicing reporter. We applied HCS-Splice to the wild-type and mutant reporters and measured the functional change in exon 10 inclusion. To demonstrate the applicability of the method in cell-based drug discovery, HCS-Splice was used to evaluate the efficacy of an exon 10-targeting siRNA, which was able to restore the correct alternative splicing balance.

## 1. Introduction

Alternative splicing occurs in approximately 95% of human genes [1,2]. It plays a crucial role in phenotypic complexity, which increases with the extent of alternative pre-mRNA splicing [3], and in the proteome [4,5]. Thus, it is possible to produce proteins with different structures and functions or to alter mRNA localization, translation, or decay starting from a specific pre-mRNA [6]. This relevant event is a critical step in the post-transcriptional regulation of gene expression, a dynamic and flexible process mediated by different regulatory elements (such as *cis*-acting and *trans*-acting factors) [7,8,9,10] and additional molecular features, as well as RNA secondary structures [11,12] and chromatin structure [13,14], which are known to promote (*enhancers*) or inhibit (*silencers*) splicing activity to generate protein diversity. All these regulatory factors, regions, and splice sites can be affected by mutations that can cause a variation in alternative splicing isoforms. Indeed, about half of known human genetic diseases are due to splicing-impairing mutations [15].

One of the diseases directly resulting from the misregulation of alternative splicing is Frontotemporal Dementia and Parkinsonism linked to chromosome 17 (FTDP-17) (OMIM 600274) [16,17]. FTDP-17 is caused by a wide range of mutations in the *MAPT* (*Microtubule-Associated Protein Tau*) gene (17q21.31; OMIM_157140; NM_016835) [18,19,20,21,22,23,24,25,26], which encodes for tau protein. About half of these mutations can induce the aberrant alternative splicing of exon 10, affecting the ratio of tau isoforms with four (4R) repeats (with exon 10) or three (3R) repeats (without exon 10), which is about 1 in the healthy adult human brain. In contrast, in pathological conditions, this ratio increases to 2–3 [27]. This unbalanced ratio determines tau protein aggregation in NFTs, and the neuropathology described in families with FTDP-17 neurodegeneration.

Several strategies using different quantitative assays have been developed to increase the likelihood of discovering molecular therapeutics capable of modulating aberrant alternative splicing [28,29,30,31]; however, these assays have limitations regarding throughput and sensitivity.

Indeed, the identification of splicing regulators for a given alternative exon is generally very complex and laborious [32], especially when the method used is a single-output splicing reporter which measures the variation in expression levels of one of the two isoforms using biomolecular assays [33,34,35]. Nevertheless, several studies have standardized the use of two-colour fluorescent minigene reporter systems in improving the dynamic range and distinguishing changes in alternative splicing from those due to transcription or translation [36,37], thus supporting the utility of such tools. Peter Stoilov and colleagues made one of the most important contributions to the development of efficient minigene reporter systems [38,39]. They showed that bioactive compounds could modulate the splicing of a two-colour (GFP/RFP) fluorescent reporter plasmid with Microtubule-Associated Protein Tau (MAPT) exon 10 [38]. The ability of this reporter to generate two alternative mRNA isoforms encoding different fluorescent proteins from a common pre-mRNA in a mutually exclusive manner provides unprecedented sensitivity for detecting changes in alternative splicing events [32,40,41].

Here, we report the development of HCS-Splice, a versatile, valid, rapid, and robust High Content Screening (HCS) assay to functionally confirm the role of splicing mutations in human genetic disorders and to identify potential molecular compounds for their effects on alternative splicing regulation. Therefore, we combined two strategies: the two-colour (GFP/RFP) fluorescent reporter plasmid assay, and the cell-based High-Content Imaging Technology, which provides higher sensitivity than the traditional plate reader-based assay.

To mimic the pathological alteration frequently observed in FTDP-17, we introduced a dementia-related splice-switching point mutation (N279K) [42] in the MAPT exon 10 two-colour (GFP/RFP) fluorescent splicing reporter, which favours the inclusion of exon 10. We then applied image-based analysis using HCS technology. Single-cell fluorescence measurement allows the calculation of multiple output parameters from a single well, providing reliable data even from small numbers of cells for kinetic assessments of alternative splicing. On a global scale, this work provides the basis for potential screening methods, useful for drugs and small molecule discovery, that can be easily miniaturised to run screening in multiwell-plate formats ranging from 96 to 1536 wells.

## 2. Materials and Methods

The experiments described do not require IRB approval.

### 2.1. Mutagenesis of Two-Fluorescent Reporter Plasmid Bearing Exon 10

The two-colour (GFP/RFP) fluorescent reporter plasmids created by Peter Stoilov and colleagues [38,39] PFLARE 5A MAPT-Exon 10 (PFLARE 5A-Tau10 WT) were used as the template for generating the resulting plasmid PFLARE 5A MAPT-Exon 10 mutant (PFLARE 5A-Tau10 mut). The QuickChange II XL Site-directed Mutagenesis Kit (Agilent Technologies, Milan, Italy) was used to introduce exon 10 N279K mutation (Chr17: 44,087,690 T > G), according to the manufacturer’s protocol.

In brief, the PCR amplification was performed in a final volume of 50 μL in a reaction mixture containing, as a template, 20 ng of PFLARE 5A-Tau 10 WT plasmid and the following specific primers: mut_Exon10-For 5′-CCAAAGGTGCAGATAATTAAGAAG-3′ and mut_Exon10-Rev 5′-GTTGCTAAGATCCAGCTTCTTCTT-3′.

The sequences of both wild-type and mutant reporters and the exact point mutation position in the PFLARE 5A-Tau10 mut plasmid were verified by sequencing (BMR Genomics, Padova, Italy).

### 2.2. SH-SY5Y Human Cell Line Culture

Human dopaminergic neuroblastoma SH-SY5Y cells (ATCC, American Type Culture Collection, Manassas, VA, USA) are a thrice-cloned (SK-N-SH -> SH-SY -> SH-SY5 -> SH-SY5Y) subline of the neuroblastoma cell line SK-N-SH. The SH-SY5Y cells were grown in a 1:1 mixture of Eagle’s Minimum Essential Medium and F12 Medium (Gibco, Life Technologies, Monza, Italy) supplemented with 2 mM L-Glutamine (Gibco, Life Technologies, Monza, Italy), 100 U/µL Penicillin/Streptomycin (Gibco, Life Technologies), and 10% Fetal Bovine Serum (FBS) (Gibco, Life Technologies, Monza, Italy). All cell cultures were grown in a humidified atmosphere at 37 °C and 5% CO_2_. Experimentally, the SH-SY5Y cells have the advantage of combining the proliferative potential of an immortal cancer cell line with the ability to differentiate into neuron-like cells that can then be used in functional assays. In addition, SH-SY5Y both undifferentiated and differentiated has been reported to express all major human tau isoforms and have efficient axonal targeting for the tau protein [43,44,45]. Therefore, all the properties of SH-SY5Y-deived neurons make them a powerful neuronal cell model for investigating the HCS-Splice method.

### 2.3. SH-SY5Y: Transfection and Cotransfection

The day before transfection or cotransfection experiments, 5 × 10^4^ cells were seeded onto 24-multiwell plates (Corning, NY, USA) and grown in 0.5 mL supplemented medium without antibiotics at 37 °C and 5% CO_2_. An amount of 0.5 μg each of PFLARE 5A-Tau10 WT or PFLARE 5A-Tau10 mut plasmid was employed in transfection or cotransfection experiments. The transfection experiments were performed according to the manufacturer’s instructions (Life-Technologies, Carlsbad, CA, USA) with a ratio of DNA (ng): Lipofectamine 3000 (μL) of 1:2 in Opti-MEM medium (Gibco, Life Technologies, Monza, Italy), and cells’ confluence of maximum 70%. The cotransfection assays were carried out using siRNA-Tau10 and siRNA Scramble molecules (Eurofins Genomics, Milan, Italy) at different concentrations, 10 nM, 25 nM, 50 nM, and 100 nM, using a ratio of DNA (ng): Lipofectamine 3000 (μL) of 1:2 in Opti-MEM medium (Gibco, Life Technologies, Monza, Italy), and cells’ confluence of maximum 70%. The control cells were treated with medium non-transfected (NT), Lipofectamine 3000 only (Mock), WT, or mutant two-fluorescent reporter plasmid and cotransfected with non-targeting siRNA (siRNA Scramble). Following transfection and cotransfection, the cells were incubated at 37 °C in 5% CO_2_ for 48 h.

### 2.4. High Content Imaging Acquisition

At 48 h of transfection or cotransfection, the fluorescent signals from each plasmid (PFLARE 5A-Tau10 WT and PFLARE 5A-Tau10 mut) were analysed on live cells using the Operetta High-content Imaging System (PerkinElmer, Monza, Italy). The SH-SY5Y live cells were washed with PBS and incubated with 1 mg/mL of Hoechst33342 fluorescent dye (ThermoFisher Scientific, Waltham, MA, USA), added to 0.5 mL of supplemented DMEM (Gibco, Life Technologies, Monza, Italy) for 20 min at 37 °C. This step was needed to enable nuclei counterstaining and subsequent total analysis. At the end of the incubation, the cells were washed and DMEM medium without phenol red was supplemented to each well (Gibco, Life Technologies, Monza, Italy). The total cell numbers, transfection efficiency, and fluorescence intensity measurements were estimated using a 20X LWD objective. The acquisition step was performed under 5% CO_2_ and 37 °C conditions in a wide-field mode in combination with three different filter combinations. The Hoechst33342 was detected with a dedicated combination of filters (excitation: 360–400 nm, emission: 410–480 nm), while the filter settings suitable for Alexa Fluor 488 (GFP: excitation: 460–490 nm, emission: 500–550 nm) and Alexa Fluor 594 (RFP: excitation: 560–630 nm, emission: 580–620 nm) were used to measure both the PFLARE 5A-Tau10 WT and PFLARE 5A-Tau10 mut reporters’ fluorescent intensities.

A minimum of 12 images per well was acquired to score enough objects for each tested condition.

### 2.5. High Content Imaging System Analysis

Harmony High Content Analysis software 4.1 (PerkinElmer, Monza, Italy) was used for the image analysis. The “Select Population” feature of the software allowed us to set a threshold of fluorescence intensity to identify the sub-population of transfected cells. The SH-SY5Y cells’ transfection efficiency analysis was carried out, taking into consideration the number of both GFP and RFP positive cells, compared to the Hoechst33342 positive total number of cells. At the same time, the two-colour (GFP/RFP) fluorescent intensity ratio of the cells was measured to evaluate the differential expression values between the PFLARE 5A-Tau10 WT and PFLARE 5A-Tau10 mut reporters’ fluorescent plasmids, and therefore between cell populations treated and untreated with siRNAs.

In brief, the live cell images acquired were segmented and analyzed using Harmony Software 4.1, according to the following workflow.

In the first step, the cells were identified using the most suitable algorithm based on nuclei segmentation, performed on the Hoechst33342 channel using the Find Nuclei building block. In the second image analysis step, assuming the homogeneous distribution of the GFP and RFP fluorescent signals in the live cells, the respective intensities were measured within a region of interest (ROI) around the nuclei, identified through a Select Region block. In order to quantify the mean fluorescence intensity corresponding to the true transfected live cells, a double-threshold strategy to filter out the cells being either positive for G (GFP) or R (RFP) or the double G + R “Yellow” fluorescent signal was applied. A Select Population function was employed to classify each fluorescent cell population, G, R, and G + R. To define the splitting point between the populations, an intensity threshold was determined based on the signal coming out from positive and negative control cells (not transfected wells). The resulting Intensity Threshold was used as a reference value to classify the three fluorescent cell subpopulations. Considering the transfected cells only, using the Find Results section, the ratio between G and R mean fluorescence intensity on a per-cell basis was calculated.

The cell subpopulations were then classified as preferentially expressing GFP (G/ R> J), RFP (G/R < K), or both the reporters G+R (K < G/R < J), by applying appropriate thresholds (J, K) for the three different subpopulations. Indeed, the lower threshold (K) and the upper threshold (J) values allowed the generation of a scatter plot of individual cells’ distribution related to the two-colour (GFP or RFP) fluorescent signals. Feature outputs of the analysis protocol included the total cell count, percentage of transfected cells calculated over the total number of cells, the G/R median value of all the cells, and the percentage of cells preferentially expressing GFP (G/R > J), RFP (G/R < K) or both the reporters (K < G/R < J) (Appendix A). Background fluorescence signals were calculated around the cell-free area and subtracted for all objects (cytoplasm-GFP, -RFP and -MERGE cells). Texture property building blocks were used. Batch analyses were carried out simultaneously for all biological experiments. 

### 2.6. Validation of HCS-Splice: RNA Extraction and Semiquantitative RT-PCR Analysis

At the end of the HCS-Splice analysis, RNA was extracted using Trizol Reagent (Life Technologies, Carlsbad, CA, USA), according to the manufacturer’s instructions. RNA was treated with DNase (TURBO DNA-free Kit; Life Technologies, Carlsbad, CA, USA), and concentrations were estimated using a Nano-Drop ND-1000 Spectrophotometer (Nano-Drop Technologies, Wilmington, NC, USA). Five hundred nanograms of extracted RNA was reverse transcribed into cDNA using dT18 oligonucleotides and the RevertAid First Strand cDNA Synthesis Kit, according to the manufacturer’s protocols (Thermo Scientific, Waltham, MA, USA). Semiquantitative RT-PCR reactions, to evaluate the expression of exon 10, were carried out in 25 μL final volume in a reaction mixture containing 50 ng of cDNA and 1U Taq DNA polymerase (Applied Biosystems by Life Technologies, Waltham, MA USA), according to the manufacturer’s protocols. After 5 min of denaturation at 95 °C, amplification was carried out for 30 cycles (30 s at 95 °C, 40 s at 60 °C, and 1 min at 72 °C), with a final extension for 7 minat 72 °C. Endogenous MAPT expression levels were detected using a pair of specific primers, at a final concentration of 1 µM, reported below: human MAPT: TAU-Ex9 For 5′-CTGAAGCACCAGCCGGGAGG-3′; TAU-Ex13 Rev: 5′TGGTCTGTCTTGGCTTTGGC-3′. The primers discriminated between the two alternative endogenous MAPT transcripts (E10+ and E10-), producing two different amplicons: 275 bp for the isoform without E10 (E10-), and 368 bp for the isoform with E10 (E10+).

To analyze the transcripts coming from the PFLARE 5A-Tau10 WT and PFLARE 5A-Tau10 mut minigene reports semiquantitative RT-PCR was performed using a final concentration of 0.5 µM of each primer: Exon 1 Bgl For 5′-AAACAGATCTACCATTGGTGCACCTGACTCC-3′, and EGFP Rev 5′-CGTCGCCGTCCAGCTCGACCAG-3′. An amplicon of 207 bp for the isoform without E10 (E10-) and an amplicon of 300 bp for the isoform with E10 (E10+) were obtained.

An amplicon of 650 bp from the Beta Actin region was amplified using primers Beta-Actin For 5′-AGACGGGGTCACCCACACTGTGCCCATCTA-3′, and Beta-Actin Rev 5′-CTAGAAGCATTTGCGGTGCACGATGGAGGG-3′ was used as a control (housekeeping gene). The final concentration used for each primer was 0.5 µM. Amplified products were loaded in 2% agarose gel electrophoresis performed in TAE 1X running buffer and visualized with Atlas ClearSight (5%) (Bioatlas, Tartu, Estonia). Then, 1 kb plus DNA ladder (Invitrogen, ThermoFisher Scientific, Milan, Italy) were run with amplified products to keep track of the band dimension. Densitometric analyses were carried out with Image Lab 2.0 software (Bio-Rad, Hercules, CA, USA).

### 2.7. Statistical Analysis

Data from three independent biological replicates, with three technical replicates for each, were represented as mean ± SD (n = 3). Data were compared using Student’s t-test and one-way analysis of variance (ANOVA), followed by Bonferroni’s multiple comparisons tests (GraphPad Software 8, San Diego, CA, USA, www.graphpad.com (accessed on 12 December 2021). Imaging batch analyses were carried out simultaneously for all biological experiments, and the Z’ Prime statistical method was applied to monitor the assay accuracy and sensitivity. The Intensity Threshold was defined by using the Z formula:(1)$$Z^{’}=1-\frac{3*\left(\sigma \;Negative\;Control+\sigma \;Positive\;Control\right)}{\;\left(µ\;Negative\;Control-µ\;Positive\;Control\right)}=1-\frac{3}{SNR}$$
where $SNR=\frac{\left(µ\;Negative\;Control+µ\;Positive\;Control\right)}{\;\left(\sigma \;Negative\;Control-\sigma \;Positive\;Control\right)}$

and µ = mean; σ = Standard deviation; *SNR*: Signal to Noise Ratio.

Z-values above ~0.4 were considered sufficient for screening assay, and Z’ values above ~0.6 were considered robust values. The theoretical maximum Z’ value was 1.0 (= no noise); however, HCS assays with a Z′ factor of 0.5 indicate that the assay was robust enough to identify molecules’ activity reliably [46]. Statistical significance was set to *p* < 0.05 and it is denoted with asterisks (* *p* < 0.05; ** *p* < 0.01; *** *p* < 0.001; **** *p* < 0.0001).

## 3. Results

### 3.1. HCS-Splice Workflow

A new image-based analysis method, HCS-Splice, was optimised to expand our ability to find new molecular therapies to modulate aberrant alternative splicing. We developed our analysis method using PerkinElmer High-Content Screening (HCS) and High-Content Analysis (HCA) technologies. Indeed, the automated combination of image acquisition (Operetta System, PerkinElmer, Monza, Italy) and analysis (Harmony Software 4.1) made it possible to extract quantitative multi-parametric data from cellular samples even at the single-cell level, allowing several questions to be answered simultaneously.

Considering the time, cost, and process optimisation of screening, the multi-step HCS-Splice workflow can be summarised in four essential canonical phases leading to drug validation in in vitro systems. These steps include target and hit identification (in silico), screening in cell system models (in vitro), data acquisition and image analysis in the building block modality (in silico), and hits identification and validation (wet lab and in silico processes) (Figure 1A). The HCS-Splice method designed could therefore be used, in vitro, for numerous applications in different areas of drug or molecular therapeutic discovery, such as oncology and cardiovascular and neurodegenerative diseases, where we need to analyse the most complex cellular models, reliably discriminate phenotypes, and turn biological data into knowledge.

### 3.2. Exon 10 Site-Directed Mutagenesis of a Two-Colour (GFP/RFP) Fluorescent Reporter

We implemented the HCS method using a two-colour (GFP/RFP) fluorescent reporter plasmid (PFlare5A-Tau10 WT) generated by Peter Stoilov and colleagues (Figure 1B) [43]. To recapitulate the splicing mutation (N279K) described in Frontotemporal Dementia and Parkinsonism linked to chromosome 17 (FTDP-17), we introduced a T > G point mutation in the exon 10 sequence, using site-directed mutagenesis (Figure 1C). Indeed, under pathological conditions, the presence of the N279K point mutation in the human *TAU* gene sequence affected splicing, allowing exon 10 to be more frequently incorporated into tau transcripts, thus increasing the expression and production of the tau isoform protein 4R compared to the 3R [19,47,48]. The presence of the N279K mutation in the PFlare5A-Tau10 mutant (PFlare5A-Tau10 mut) fluorescent reporter plasmid compared to the PFlare5A-Tau10 wild-type (PFlare5A-Tau10 WT) sequence was confirmed with Sanger sequencing (Appendix A).

### 3.3. Development of an Image-Based HCS-Splice Method Using Two-Colour (GFP/RFP) Fluorescent Reporter Plasmids Transfected into SH-SY5Y Cells

To assess whether the PFlare5A-Tau10 mut fluorescent reporter plasmid was able to recapitulate the aberrant alternative splicing of exon 10 in the presence of the N279K mutation and whether it was suitable to implement our new HCS-Splice method, we transiently transfected either PFlare5A-Tau10 WT or PFlare5A-Tau10 mut into Human dopaminergic neuroblastoma SH-SY5Y cells (Figure 1A and Figure 2A). These cells presented splicing *trans-acting* factors that allow for the inclusion of exon 10 in 30% of endogenous MAPT transcripts (Appendix A).

After 48 h of transfection, images of the SH-SY5Y cells were acquired and analysed using a high content screening method, and cell viability was first measured (Appendix A).

Therefore, by using the high content imaging through the automated fluorescent microscope (Operetta, PerkinElmer, Monza, Italy), images were acquired from multi-well plates, in live cells condition, under CO_2_ controlled (5%) and at a temperature of 37 °C, using a 20X LWD objective in wide-field mode, with the following filter combinations for Hoechst33342 (excitation: 360–400 nm; emission: 410–480 nm), Alexa Fluor 488 (excitation: 460–490 nm; emission: 500–550 nm), and Alexa Fluor 594 (excitation: 560–630 nm; emission: 580–620 nm). During the automated image acquisition, the system performed a fully automated measurement of the different fluorescent intensity signals, Hoechst, RFP, and GFP expressed by transfected SH-SY5Y cells (Figure 2A).

The analysis workflow was optimized and used to quantify outcome signals of PFlare5A-Tau10 mut compared to the PFlare5A-Tau10 WT fluorescent reporter plasmid.

A brief flowchart of the image analysis sequence is shown in Figure 2B: at first, the pixel gradient of the Hoechst image was segmented to define the individual nucleus of every single cell. To assess the ability to detect the cytoplasmic GFP, RFP, and MERGE regions at the single cell level, and after clustering at the population level, the fluorescence intensity signals for both Alexa Fluor 488 and 594 channels were defined around the nuclear region. The fluorescence intensities were measured both within the cells and in the background region (cell-free area). The background-corrected fluorescence intensity was used as a readout to define the objects of interest, the “True cell population”.

Subsequently, the select population building block allowed us to classify three subpopulations from the “True cell population” related to the different fluorescence signals. The machine learning algorithms based on two proprieties, Filter by Property and Linear Classifier, selected the key parameters necessary to set thresholds based on the intensities of RFP and GFP fluorescence of the objects of interest (Appendix A).

This step allowed the generation of a scatter plot showing the distribution of individual cells in each well carrying one of the two-colour (GFP or RFP) fluorescent reporters (PFlare5A-Tau10 WT or Pflare5A-Tau10 mut) (Figure 3A). The thresholds values, K = 2 (lower threshold) and J = 5 (upper threshold), were chosen based on the scatter plots, and three different subpopulations were identified: (1) the GFP-positive cells, which had a G/R parameter higher than 5 (J threshold); (2) the cells that predominantly expressed RFP, which had a G/R lower than 2 (K threshold); (3) the cells in which RFP and GFP were expressed in roughly the same amounts, which had a G/R parameter between 2 and 5 (K < G/R < J) (Figure 3A).

This single-cell classification allowed us to follow the splicing of MAPT exon 10 since the GFP population corresponds to cells in which the majority of report transcripts are devoid of exon 10, the RFP population corresponds to cells in which the majority of reporter transcripts contain exon 10, and the “Yellow” population, which shows equal levels of GFP and RFP fluorescence, reports both splicing variants. Indeed, the selected read-outs of the quantitative analysis of the two reporter plasmids showed that the presence of the N279K point mutation in the PFlare5A-Tau10 mut altered the fluorescence signals from a relatively low level of RFP to a very high level due to the inclusion of exon 10 (the characteristic hallmark of FTDP-17) (Figure 2A and Figure 3A,B).

In particular, SH-SY5Y cells transfected with PFlare5A-Tau10 mut compared to the PFlare5A-Tau10 WT fluorescent reporter plasmid showed, as expected, a shift in the proportions of the three sub-populations expressing either RFP-E10+, GFP-E10- or both “Yellow- E10+/E10-”. The RFP cells increased from ~5% RFP to ~58% of the total cells, while the GFP cells decreased from ~10% GFP to ~5%. The population expressing both E10+ and E10- at the same time, reporting the transcript “Yellow”, simultaneously decreased from ~82% YFP to ~43% (Figure 3B).

### 3.4. HCS-Splice Data Validated Using Semi-Quantitative RT-PCR

Semi-quantitative RT-PCR was performed, as described in Materials and Methods, to validate HCS-Splice image-based results (Figure 3C). As schematically shown in Figure 1B, the alternative splicing of exon 10 resulted in the production of two different splice isoforms (E10+ and E10-). The presence of exon 10 in the reporter PFlare5A-Tau10 transcript resulted in a 300 bp amplicon (E10+), whereas the amplification of the reporter transcript without E10 resulted in a 207 bp amplicon (E10-). As the reverse primer annealed to the reporter-specific portion of the RNA transcribed from the GFP cassette (Materials and Methods Section), the endogenous tau mRNA was not amplified by this primer pair.

Densitometric analyses of each band relative to the two different isoforms (E10+ and E10-), normalized using the housekeeping gene Beta Actin (650 bp), showed that the level of exon 10 transcript (E10+) content increased from ~35% in WT to ~85% in the mutant reporter plasmid (*p* < 0.001), in accordance with the increase in RFP signal observed with image analysis (Figure 3B).

At the RNA level, we observed that the N279K mutation increased the exon 10 levels by about 2.5-fold compared with the PFlare5A-Tau10 WT (Figure 3C), as reported in FTDP-17 patients [49,50]. We were able to understand that the level of endogenous exon 10 mRNA expression was comparable to levels observed by transfecting the reporter PFlare5A-Tau10 WT (~35%). Therefore, to improve the HCS-Splice methods to monitor the slight variations of splicing induced by potential therapeutic compounds during screening in vitro, we used our two-colour PFlare5A-Tau10 mut fluorescent reporter plasmid.

### 3.5. HCS-Splice Method Validation by Using the siRNA Molecules Modulating the Exon 10 Splicing

To investigate the potential of our HCS-Splice method to effectively identify drugs or molecules that modulate splicing, a siRNA molecule targeting MAPT exon 10 (siRNA-Tau10), and a control siRNA (siRNA-Scramble) were designed. Wild-type and mutant two-fluorescent plasmids (PFlare5A-Tau10 mut and PFlare5A-Tau10 WT) (Figure 1B) and siRNAs (siRNA-Tau10 and siRNA Scramble) were transiently co-transfected into SH-SY5Y cells in 24-wells plates. This plate format was chosen to allow the recovery of a sufficient number of cells at the end of the assay after the image-based analysis, for the downstream processing for molecular analysis, such as semi-quantitative RT-PCR. The seeding density of cells can significantly affect high-content analysis. Therefore, it is recommended that a good balance be achieved between a high density for statistically meaningful results and a low density for robust cell detection.

The efficacy of siRNA-Tau10, designed to silence exon 10-containing transcripts, was analysed in HCS-Splice at different concentrations ranging from 10 nM to 100 nM, in co-transfection with 0.5 μg of PFlare5A-Tau10mut (Figure 4).

A dose-dependent reduction of the exon 10 isoform, E10+, was observed in PFlare5A-Tau10mut-transfected SH-SY5Y cells treated with siRNA-Tau10, relative to controls (Figure 4A). While the percentage of red cells (RFP-E10+ isoform) decreased, there was an increase in the percentage of green cells, indicating the selective degradation of the alternative transcripts containing exon 10 (RFP-E10+). The treatment with 10 nM of siRNA-Tau10 resulted in a slight reduction of the red population by ~5% compared to the scramble control (*p* < 0.05). A concentration of 25 nM reduced red cells by ~15% (*p* < 0.05). Concentrations of 50 nM and 100 nM led to reductions of ~ 25% (*p* ≤ 0.01) and ~ 45% (*p* ≤ 0.001), respectively. This gradual decrease in red cells was accompanied by a specular increase in green cells compared to the scramble control (5%, 5%, 7%, and 35%, respectively). Furthermore, we observed a slight increase of ~ 5–10% in yellow cells, compared to the scramble control (Figure 4B).

In a cotransfection experiment in SH-SY5Y cells with PFlare5A-Tau10 WT and siRNA-Tau10 (50 nM and 100 nM, respectively), we observed an increase in the percentage of green cells (Appendix A) and a decrease in yellow cells. However, when transfected with the WT reporter, a lower proportion of cells expressed only RFP in comparison with cells transfected with the mut reporter. Therefore, the assay showed a limited dynamic range for detecting exon 10 reduction in the context of PFlare5A-Tau10 WT (Appendix A).

### 3.6. Semiquantitative RT-PCR Analysis to Validate the Effects of the siRNA after HCS-Splice Assay

As a final step, a semiquantitative RT-PCR assay was carried out to validate the data obtained with the HCS-Splice image-based assay. Total RNA was extracted from the SH-SY5Y cell samples previously described and analysed. Following the cDNA synthesis, semiquantitative RT-PCR was performed to investigate the effects of siRNA-Tau10 treatment at concentrations ranging from 10 nM to 100 nM (Figure 5).

Densitometric analysis showed that the siRNA-Tau10 induced the depletion of transcripts containing exon 10 (E10+) in a dose-dependent manner, confirming the results obtained with the HCS-Splice methods. Indeed, each band relative to the two different isoforms (E10+ and E10-), normalized using the housekeeping gene Beta (β) Actin (650 bp), showed that upon treatment with 10 nM there was no significant difference between the level of E10+ compared to the siRNA scramble control (*p* = n.s.), whereas, at a concentration of 25 nM, we observed a slight decrease in E10+, by ~5% (*p* < 0.05) (Figure 5A). Interestingly, we were able to gradually decrease the level of E10+ by almost ~10% (*p* ≤ 0.01) after treatment with 50 nM and by ~35% (*p* ≤ 0.001) by using the siRNA-Tau10 at a concentration of 100 nM (Figure 5B). The control siRNA scramble at different concentrations did not affect the E10+ isoform, as expected (Figure 5). The untreated mutant plasmid was compared with the different siRNA treatment conditions. These data further support and confirm the effects observed through the HCS-Splice image-based analysis.

## 4. Discussion

Here, we describe the new HCS-Splice method assay to allow the identification of molecules to modulate splicing outcomes.

Neuroblastoma cell lines are still not easy to transfect. By combining the two-colour fluorescent reporter, transiently transfected into the SH-SY5Y cell line, with high-content image-based screening, we were able to restrict the analysis to only a subset of transfected cells, providing a powerful method for detecting rare events and even individual cells. The plate-reader-based assay originally proposed by Stoilov [38] involves a cell lysis passage and the recording of two fluorescence intensity values resulting from the entire cell population in each well, resulting in a single ratiometric readout (GFP/RFP) [43]. Despite the robustness and suitability of the traditional plate-reader-based assay for high throughput screening, as reported in the literature, we believe that the HCS-Splice method reported here can take advantage of a per-cell-based image analysis approach that allows the recording of additional data such as the number of cells per well, the shape of both nuclei and cytoplasm, single cell morphology, viability, and transfection efficiency, which are helpful to monitor cytotoxicity due to the effects of compounds used in the study.

As part of the development of the minigene model for FTDP-17, we introduced the N279K mutation into the PFlare5A-Tau10 WT two-colour (GFP/RFP) fluorescent reporter plasmid developed by Stoilov and colleagues [38,39], thereby increasing red cells from ~5% to ~58% and decreasing green cells from ~10% to ~5%. This change was associated with a modification in the third subpopulation of cells containing both isoforms, exon 10+ and exon 10- (“Yellow cells”), which decreased from ~82% to ~43%. This decrease reflects a conversion of most cells to favour the production of exon 10+ in the PFlare5A-Tau10 mut reporter plasmid (Figure 3B).

The HCS-Splice method results and the semiquantitative RT-PCR assay data are not immediately translatable, one into the other, since one method is an image-based assay measuring proteins, while the other measures mRNA isoforms. In particular, the image-based method was applied to three subsets of populations (RFP-E10+, GFP-E10-, and Yellow-E10+/E10-), whereas the RT-PCR detected two different mRNA isoforms (RFP-E10+ and GFP-E10-).

Despite these disparities, we validated the HCS-Splice results via semiquantitative RT-PCR: the transcript containing exon 10 (E10+) increased from about 35% of the PFlare5A-Tau10 WT reporter plasmid to about 85% of the PFlare5A-Tau10 mut reporter plasmid (Figure 3C).

Similarly, when the PFlare5A-Tau10 mut-transfected cells were treated with an effective compound at increasing concentrations, the increase in green cells’ percentages was recapitulated by increased percentages of exon-10- amplicons in semiquantitative RT-PCR.

The 24-well plates were chosen to recover sufficient cells at the end of the image-based analysis assay to perform downstream molecular analyses, such as semiquantitative RT-PCR. Cell seeding density can have a significant impact on high-content analysis. Therefore, a good balance between a high density for statistically meaningful results and a low density for robust cell detection is recommended.

We tested the functionality of the PFlare5A-Tau10 mut reporter plasmid and of the HCS-Splice assay system using several splice modulators [51]. An example of one such modulator has been shown here to reduce exon 10 (E10+)-containing transcripts (Figure 4). The alteration can be envisaged as a therapeutic approach that allows the cells to become similar to their wild-type counterparts.

## 5. Conclusions

Our results confirm that the HCS-Splice method, based on a two-colour (GFP/RFP) fluorescent reporter plasmid, can be used to quantify the splicing modulation signal in vitro with greater robustness than existing methods. The integrated HCS-Splice method will have several applications. Firstly, it will broaden the use of two-colour fluorescent reporters to validate putative splicing mutations in human diseases, thereby improving the ability to screen potential splicing-modulating candidates based on nucleic acids, such as ASO or siRNA, able to restore the correct alternative splicing balance. Furthermore, due to the easy miniaturisation of cell culture plate formats, ranging from 96 to 1536 wells, the proposed HCS-Splice method could be applied on a global scale in the screening of both drug and small therapeutic RNA molecule libraries in the context of high throughput screening.

## Figures and Tables

**Figure 1 cells-12-01959-f001:**
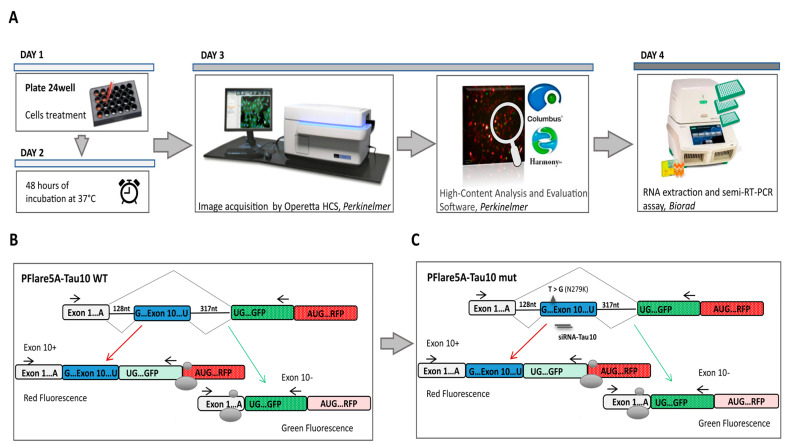
Setting up a high content imaging cell-based assay. (**A**) Workflow for the screening of molecular compounds. After transfection (Day 1) and incubation time (48 h of treatment) (Day 2), the images are acquired from the 24-well plates (Day 3) and analysed using the high-content imaging System. To validate data observed through image-based assay, the samples were subjected to downstream analyses, such as semiquantitative RT-PCR. (**B**) Schematic representation of the two-fluorescence reporter plasmid (not to scale). Alternative splicing of exon 10 leads to the production of two different splice isoforms from the two-fluorescent reporter plasmid (PFlare5A-Tau 10). The ribosome scans for the first available AUG codon to initiate the translation of specific reporter proteins such that exon 10+ results in the production of RFP and exon 10- results in the production of GFP. (**C**) The position of the N279K mutation, which leads the increased RFP production due to exon 10 inclusion, is indicated by a triangle. siRNA TAU 10 site-position was reported. The horizontal arrows, in (**B**,**C**), represent primers for semiquantitative RT-PCR analysis. The endogenous TAU mRNA is not amplified by this primer pair, as the forward and reverse primer anneals to the reporter-specific portion of the RNA transcribed from the GFP sequence cassette.

**Figure 2 cells-12-01959-f002:**
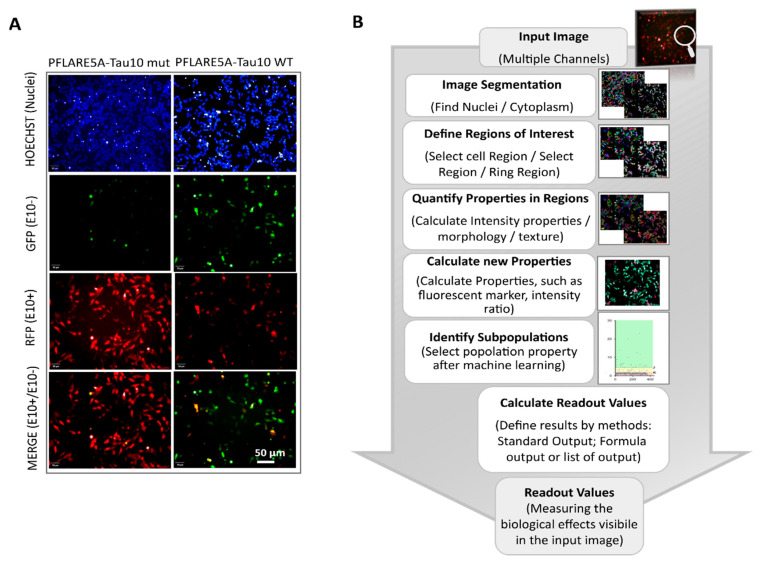
HCS-Splice analysis protocol applied for in vitro splicing effects measurement (Harmony Software 4.1, *PerkinElmer*). (**A**) Representative images of SH-SY5Y cells 48 h after transfection with the wild-type (left panel) and Mutant N279K PFlare5A-Tau10 reporter plasmids (right panel). The images were acquired using a 20X LWD objective, in a wide-field mode in combination with filters for Hoechst33342 (excitation: 360–400 nm; emission: 410–480 nm), Alexa Fluor 594 (excitation: 560–630 nm; emission: 580–620 nm), and Alexa Fluor 488 (excitation: 460–490 nm; emission: 500–550 nm), using the automated fluorescence microscope platform (Operetta High Content Image System, *PerkinElmer*). The scale bar represents 50 µm. (**B**) Steps of image analysis were performed after 48 h of transfection with Harmony software 4.1 (*PerkinElmer*) in a stepwise manner: Hoechst staining was used for nuclei segmentation and the detection of ROIs; nuclear identification was followed by the automated definition of the cytoplasmic regions of interest; texture proprieties were used to identify the cell-free area, and fluorescence intensities were used as background for all objects (cytoplasm-GFP, -RFP, and -MERGE cells). The single object, as well as average per well fluorescence intensity, was calculated per each classified object (Appendix A).

**Figure 3 cells-12-01959-f003:**
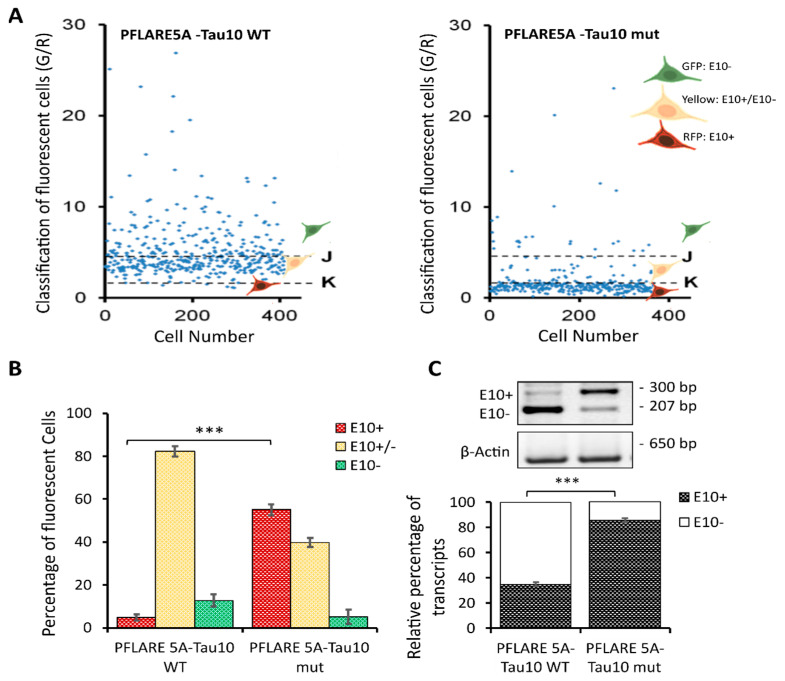
HCS-Splice to quantify the functional alteration of MAPT exon 10 inclusion. (**A**) Scatter plots representing green and red values of every single cell in a particular well bearing one of the PFlare5A-Tau10 dual reporter plasmids, wild-type or mutant, respectively. The indicated threshold values were selected to classify the cell population into three classes (E10+, E10-, and E10+/E10-) based on the intensity ratio of GFP/RFP. Representative scatter plots of independent experiments (n = 3) are shown. (**B**) The histogram represents the relative percentages of the three sub-populations of cells (RFP-E10+, GFP-E10-, and Yellow-E10+/E10-) classified by fluorescence intensity in cells transfected with the wild-type and mutant reporter plasmids. The two red bars represent cells below the K threshold in (**A**); therefore, cells are predominately red (E10+). The yellow bars represent the cells between J and K thresholds in (**A**), and therefore cells that are predominately yellow are E10+/E10-, and the two green bars represent cells above the J threshold in (**A**), therefore cells that are predominately green are E10-. The values represent average read-outs of three biological experiments (mean ± SD). (**C**) Semiquantitative RT-PCR analysis of reporter transcript from SH-SY5Y cells transfected with the wild-type (PFlare5A-Tau10 WT) or mutant reporter (PFlare5A-Tau10 mut) plasmids. The alternative splicing of exon 10 leads to the production of two different splice isoforms (E10+ and E10-). As shown in the representative picture of gels electrophoresis, the presence of exon 10 in the reporter transcript yields a 300-bp-long amplicon (E10+), while the transcript devoid of Exon10 produces a 207-bp-long amplicon (E10-). The histogram represents the densitometric analysis of the bands relative to the two different isoforms (E10+ and E10-) normalized to β actin (*p* < 0.001). The analysis has been performed in triplicate (mean ± SD). Asterisks (*) indicate significant differences (*t*-test, *** *p* ≤ 0.001).

**Figure 4 cells-12-01959-f004:**
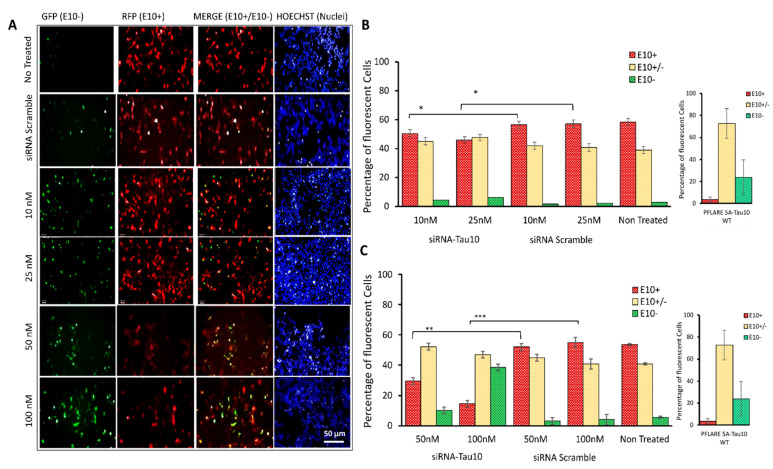
HCS-Splice to evaluate the efficiency of an Exon 10-Target siRNA for splicing modulation. (**A**) Representative images of SH-SY5Y cells cotransfected with PFlare5A-Tau10 mutant plasmid and siRNA Scramble or Exon 10-Target siRNA at concentrations of 10 nM, 25 nM, 50, and 100 nM (top to bottom). Images were acquired with a high-content screening system (Operetta, *PerkinElmer*) 48 h after transfection, as described in Figure 2A. The scale bar represents 50 µm. (**B**,**C**) The histogram represents the relative percentages of the three sub-populations (E10+, E10+/-, and E10-) of cells containing the mutant reporter (PFlare5A-Tau10 mut) plasmids treated with (**B**) 10 nM and 25 nM and (**C**) 50 nM and 100 nM of Exon 10-Target siRNA (siRNA-Tau10). Asterisks (*) indicate significant differences (*t*-test, * *p* ≤ 0.05, ** *p* ≤ 0.01, *** *p* ≤ 0.001).

**Figure 5 cells-12-01959-f005:**
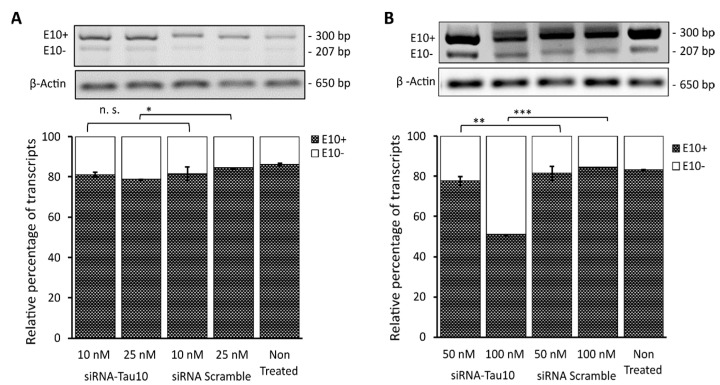
Semiquantitative RT-PCR analysis reporter transcript in SH-SY5Y cells transfected with PFlare5A-Tau10 mutant plasmid. The representative gels electrophoresis shows the semiquantitative RT-PCR products of transcripts containing exon 10+ (300 bp) and exon 10- (207 bp) upon treatment with siRNA Scramble or Exon 10-Target siRNA at (**A**) concentrations of 10 nM and 25 nM and (**B**) at concentrations of 50 nM and 100 nM. The histogram represents the percentage of relative expression of both isoforms’ exon 10+ and exon 10- for each treatment condition normalized on β actin, used as a housekeeping gene (650 bp). The analysis was performed in triplicate (mean ± SD). Asterisks (*) indicate significant differences (*t*-test, n.s *p* > 0.05, * *p* ≤ 0.05, ** *p* ≤ 0.01, *** *p* ≤ 0.001).

## Data Availability

The primary data for this study are available from the authors upon request.

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
