# Peer review of "HCS-Splice: A High-Content Screening Method to Advance the Discovery of RNA Splicing-Modulating Therapeutics"

_cells, 2023, doi:10.3390/cells12151959_

Round 1

Reviewer 1 Report

The manuscript describes the development of a versatile, effective, rapid, and robust high-throughput strategy for the identification of potent splicing-modulating molecules by combining the the use of a two-colour (GFP/RFP) fluorescent splicing reporter plasmid and cell-based High-Content Screening (HCS) assay. The name of the proposed approach is HCS-Splice. Authors suggest its exploitation to functionally confirm splicing mutations in human genetic disorders, to screen drug candidates and, most importantly, for the identification of potent splicing-modulating molecules. Authors applied the HCS-Splice to evaluate splicing changes in the presence of a Microtubule-Associated Protein Tau (MAPT) exon 10 splicing mutation and to evaluate the efficacy of an exon 10-targeting siRNA.

Overall, albeit the experimental setup has been properly conducted, the manuscript does not have the novelty required to be published in the cells journal. In fact, High-Content Screening (HCS) assay results from the combination of a two-colors (GFP/RFP) fluorescent splicing reporter plasmid with a high-technology, not common, Imaging System (Operetta High-content Imagin System). Therefore, authors do not develop any new splicing assay, but perhaps developed an automated acquisition procedure to screen cell-specific splicing pattern.

Another limitation of the study is related to the lack of high throughput screening of splicing-modulating drugs/RNA therapeutics, the main purpose of the developed approach.

Other critical points are:

-In figure 4 b and c, authors show that the siRNA-treatment lead to a dose-dependent reduction of the E10+ splicing isoform. Anyway, it is not clear why the E10+/- signal is not affected (yellow bars in the histograms) as well as authors do not provide any explanation of the counterintuitive observed increase of E10- signal (green bars in the histograms).

-Why authors exploited semi quantitative PCR instead of qPCR to evaluate the E10+ and E10- transcripts? While semi quantitative PCR is affected by several technical limitations that prevent accurate quantification of transcripts, splicing-specific qPCR should provide results as close as those provided by HCS assay.

Reviewer 2 Report

In the current manuscript, the authors developed a cell-based High-Content Screening (HCS) assay, called HCS-Splice, using a two-color (GFP/RFP) fluorescent splicing reporter plasmid. In addition, they introduced a mutation in the Microtubule-Associated Protein Tau that alters the splicing of exon 10. They also tested the efficacy of siRNA targeting exon 10 to restore the normal alternative splicing pathway. In general, the study is well designed, written, and presented. However, I have some comments and suggestions for the authors.

Line 35: delete “to”

Line 105: “Quick-change” should be “QuikChange”.

Line 109: How much of the plasmid template was added to the PCR mixture?

Line 124: SH-SY5Y: Transfection and Cotransfection: How about the replication of the experiments here? How many replicates were used for the control and treatment? Please add this information to the methods.

Why did the authors use the SH-SY5Y human cell line? Please provide the rationale.

Line 132: Please delete “by”.

Line 133: Please add a space between the number and measuring unit “10 nM, 25 nM”.

Line 133: Please provide more details on the transfection and cotransfection assay so that it can be easily replicated by other researchers.

Line 185: Please define J and K. They are defined in the results section (340-347) but should be defined here at the first time you use them.

Lines 201-205: Please complete the PCR conditions such as the primers used and the amount added to PCR, … etc.

Line 234: Please define SNR.

Line 261: Please delete “by”.

Please edit Figure 1A to add Day 2.

Line 318: Add a hyphen. “… background-corrected …”

Line 379: “was able to increase” should be “increased”.

Line 440: … reduced …

Line 472: … housekeeping gene …

Line 477: please add a space “50 nM”.

Line 504: …. we restricted the analysis ….

Line 673: Caenorhabditis Elegans should be italicized, Elegans should be lowercase “Caenorhabditis elegans

The authors should use more recent references. There is only 4 references from 2018-2023.For example, there is only one reference in 2023, one in 2022, zero references in 2020 and 2021, one in 2019, one in 2018.

I suggest a professional editing of the manuscript language. Some sentences are too long, a few are difficult to understand.

Reviewer 3 Report

This was  a well written manuscript and was very interesting to read.  The authors have developed a novel RNA splicing assay  using a 2 colour fluorescent reporting plasmid to monitor exon inclusion versing silencing using a target gene of medical importance. The authors have used this as a pilot study to work up the technology to evaluate a high throughput assay where they could screen drugs, chemicals for their ability to modulate splicing that could benefit the treatment of human diseases in the long run.

The introduction was long and felt that it could be shortened, but it did provide the reader with sufficient information to the background and the reason for undertaking the work.  Methods were well written, Results , figures were also well presented in a professional manner.

THe Discussion came across as being repetitve to what was presented in the Results section. It was quite a long discussion and so could be shortened  and would encourage the athors just to focus on the key aspects of the results.  Also the Conclusion was a bit of overreach by the authors in terms of the potential for future use.  The results presented in the current study were only in vitro based assays- with perfectly good reason, but the splice method would only be of true value if it could identify therapeutc targets that modulated splicing and they could demosntrate that the drugs were effective in achieving the outcome in vivo. Until that is achieved they may need to temper the enthusiasm with respect to the findings presented here. I do wish them well as it is a really nice peice of work.

one typo:

Line 442 - specular should be spectacular

Round 2

Reviewer 1 Report

The authors fulfilled all my points. I do not have any further concerns